# Implicit Bayesian Markov Decision Process for Resource-Efficient Experimental Design in Drug Discovery

## Abstract

In drug discovery, researchers make sequential decisions to schedule experiments, aiming to maximize probability of success towards drug candidates while simultaneously minimizing expected costs. However, such tasks pose significant challenges due to complex trade-offs between uncertainty reduction and allocation of constrained resources in a high-dimensional state-action space. Traditional methods based on simple rule-based heuristics or domain expertise often result in either inefficient resource utilization due to risk aversion or missed opportunities arising from reckless decisions. To address these challenges, we developed a Implicit Bayesian Markov Decision Process (IB-MDP) algorithm that constructs an implicit MDP model of the environment's dynamics by integrating historical data through a similarity-based metric, and enables effective planning by simulating future states and actions. To enhance the robustness of the decision-making process, the IB-MDP also incorporates an ensemble approach that recommends maximum likelihood actions to effectively balance the dual objectives of reducing state uncertainty and optimizing expected costs. Our experimental results demonstrate that the IB-MDP algorithm offers significant improvements over traditional rule-based methods by identifying optimal decisions that ensure more efficient use of resources in drug discovery.

## 1 Introduction

In drug discovery, strategic planning and selection of experiments play a pivotal role in impacting the pace and expenses of R&D activities. The identification of potential drug candidates requires conducting numerous assays at various stages of preclinical studies. The process often begins with limited information, creating significant challenges for achieving optimized outcomes due to time and budget constraints. Optimizing the use of resources to achieve targeted goals within these limitations is among the most demanding tasks in creating effective Research Operation Plans (ROP). Conventional appoarches, often relying on simple rule-based heuristics or domain expertise, struggle to adapt as new data emerges and typically fail to address state, model, and parameter uncertainties effectively. Consequently, this results in suboptimal decision-making and inefficient allocation of resources Puterman (2014).

To address these challenges, we propose the **Implicit Bayesian Markov Decision Process (IB-MDP)** algorithm, a *model-based* approach that constructs an implicit model of the environment's dynamics by integrating historical data through a distance-based similarity metric. Unlike traditional MDP methods that require explicit modeling of transition probabilities, the IB-MDP leverages historical data to build a flexible model of the environment without the need for precise parameterization Rainforth et al. (2024); Bellet et al. (2013). This implicit model captures complex, nonlinear relationships within the data manifold, enabling efficient planning by simulating future states and actions Alagoz et al. (2010).

Moreover, to improve the robustness and reliability of decision-making, we incorporate an ensemble approach into the IB-MDP. Ensemble methods aggregate multiple policies derived from independent algorithm runs, reducing variance and mitigating bias in policy estimation Dietterich (2000); Osband

et al. (2016); Zhou (2012). This approach ensures more stable and generalizable policies, particularly in high-dimensional and resource-constrained environments Lakshminarayanan et al. (2017).

Our algorithm is demonstrated in the context of assay scheduling and ROP optimization, where it significantly improves resource utilization and decision quality compared to traditional heuristic-based approaches. The IB-MDP framework is broadly applicable to various resource-constrained decision-making tasks in drug discovery, making it a valuable tool for optimizing sequential decisions in preclinical studies.

**Summary of Contributions**:

- We introduce the Implicit Bayesian Markov Decision Process (IB-MDP), a model-based algorithm that integrates historical data using a distance-based similarity metric within the MDP framework, enabling efficient planning in sequential decision-making tasks.

- We incorporate an ensemble approach to enhance policy estimation, providing theoretical justification for its effectiveness in variance reduction, bias mitigation, and improved generalization.

- We validate our approach through experiments in assay scheduling, demonstrating significant improvements in resource utilization and decision quality over traditional heuristic methods. However, our algorithm is broadly applicable across a range of decision-making problems.

## 2    RELATED WORK

The optimization of decision-making under uncertainty has been a central focus in various domains, including drug discovery.

**Markov Decision Processes and Model-Based Reinforcement Learning**: Markov Decision Processes (MDPs) provide a mathematical framework for modeling sequential decision-making where outcomes are partly random and partly under the control of a decision-maker (Puterman, 1994; 2014). In drug discovery, MDPs have been applied to tasks such as clinical trial optimization (Bennett & Hauser, 2013; Eghbali-Zarch et al., 2019; Abbas et al., 2007; Fard et al., 2018). However, applying MDPs to experimental scheduling has remained limited due to the difficulty of accurately specifying transition probabilities and reward functions. Model-based reinforcement learning (RL) offers an alternative by learning models of the environment to improve sample efficiency and planning accuracy (Sutton, 2018; Kaiser et al., 2019). In the drug discovery field, model-based RL has been used for molecule generation (Wang et al., 2021; Bengio et al., 2021; You et al., 2018; Zhou et al., 2019), synthesis planning (Segler et al., 2018), and experimental design (Schneider et al., 2020). These methods typically require accurate environment models, a challenge in high-dimensional and complex biological systems such as those found in preclinical studies.

**Incorporating Historical Data and Similarity Metrics**: Leveraging historical data is crucial for improving decision-making in contexts with limited experimental data. Bayesian approaches, including Bayesian reinforcement learning and optimization, maintain a posterior distribution over parameters or value functions, updating beliefs based on new data (Ghavamzadeh et al., 2015; Shahriari et al., 2015). In drug discovery, Bayesian optimization has been applied to optimize molecular properties (Griffiths & Hernández-Lobato, 2020; Gómez-Bombarelli et al., 2018), but such methods are often less effective in sequential decision-making scenarios. Using similarity metrics within MDPs can further enhance the integration of historical data. Kernel-based methods, which use similarity functions to generalize across states, have been explored in reinforcement learning to estimate transition dynamics more accurately (Ormoneit & Sen, 2002; Kveton & Theocharous, 2012; Xu et al., 2007). Our approach extends this by incorporating a variance-normalized distance metric to dynamically integrate historical data into the MDP transition function.

**Ensemble Methods in Reinforcement Learning (RL)**: Ensemble methods have gained popularity for improving the robustness and reliability of decision-making. By aggregating multiple models or policies, ensemble techniques reduce the variance and bias inherent in individual estimates (Dietterich, 2000; Osband et al., 2016; Wiering & Van Hasselt, 2008; Zhou, 2012). In RL, ensemble methods are particularly effective in improving exploration and generalization, as demonstrated by their successful application in model-based RL (Lakshminarayanan et al., 2017). Our IB-MDP

framework incorporates ensemble methods to enhance decision robustness, where multiple policies derived from independent algorithm runs are aggregated to produce more reliable decision paths.

**Applications in ADME Studies and Comparison to Existing Methods**: ADME studies focus on the absorption, distribution, metabolism, and excretion (ADME) of drugs to understand their pharmacokinetic properties and impact on effectiveness and safety (Hoffman, 1998; Hoffman et al., 2004; Hughes et al., 2011). Decision-making in ADME studies requires balancing information gain with constrained resources. Although RL has been applied to clinical trial optimization (Coronato et al., 2020; Escandell-Montero et al., 2014; Martín et al., 2020), its application to preclinical experimental scheduling is underexplored. Our IB-MDP framework addresses this gap by providing a flexible and scalable decision-making approach that integrates historical data and real-time experimental results. Unlike traditional methods, the IB-MDP does not require manual specification of transition probabilities, instead leveraging a similarity-based metric to model the environment's dynamics implicitly. This, combined with the ensemble approach, distinguishes our method from existing techniques, ensuring both adaptability and robustness in decision-making.

## 3 A Sequential Decision-Making Problem Statement

In ADME studies, a primary challenge is the optimal scheduling of multiple experimental assays that contribute to evaluating a drug's ADME profile. Critical ADME assays for central nervous system (CNS) drugs involve assessing whether the drug acts as a substrate for transporters such as P-glycoprotein (PgP) and Breast Cancer Resistance Protein (BCRP). The goal is to plan in vitro PgP and BCRP assays to maximize information gain towards the drug's brain penetration potential, which can be evaluated through in vivo unbound brain-to-plasma partition coefficient ($k_{puu}$), while minimizing operational costs and adhering to resource limitations.

The focus here is on reducing state uncertainty, which refers to the incomplete knowledge about the final target feature, such as $k_{puu}$, rather than model uncertainty. Ensuring these features fall within desirable ranges is key to determining a drug's efficacy and safety.

The problem can be formulated as finding an optimal policy $\pi^*$ that minimizes cost and reduces state uncertainty, subject to constraints ensuring the likelihood of achieving experimental outcomes. This can be expressed as: $\min_\pi \mathbb{E}_\pi \left[ \sum_{t=0}^T \gamma^t R(s_t, \pi(s_t)) \right]$ subject to:

1. State uncertainty at the terminal stage $\mathcal{H}(s_T)$ must be below a threshold $\epsilon$: $\mathcal{H}(s_T) \leq \epsilon$.
2. The likelihood of achieving desirable outcomes $\mathcal{L}(s_T)$ must exceed a minimum value $\tau$: $\mathcal{L}(s_T) \geq \tau$.
3. At each intermediate step $t$, the likelihood $\mathcal{L}(s_t)$ must also exceed the threshold $\tau$: $\mathcal{L}(s_t) \geq \tau, \quad \forall t = 0, \ldots, T-1$.

Checking the likelihood at each intermediate step ensures that the decision-making process stays aligned with the final goal, maintaining a high probability of achieving desired experimental outcomes. This dynamic constraint helps the policy continuously adapt as new data emerge, enforcing the likelihood requirement throughout the MDP decision process and preventing early decisions from compromising long-term objectives. By consistently ensuring both cost efficiency and target feature accuracy, the policy remains robust and focused until the end of the search, aligning with the principles of constraint optimization in MDP frameworks.

## 4 Implicit Bayesian Markov Decision Process (IB-MDP) for Resource-Efficient Decision Making

### 4.1 Framework Description

The IB-MDP algorithm is designed to optimize experimental scheduling in resource-constrained settings by leveraging historical data through a distance-based similarity metric. This framework aims to strategically select assays to minimize costs and maximize information gain, particularly in high-dimensional decision spaces, such as assays scheduling in preclinical pharmacokinetics and pharmacodynamics (PKPD) space.

The algorithm starts with a partially known initial state and a collection of potential experimental configurations (i.e., action sets in an MDP framework). As it explores the state-action space, the IB-MDP dynamically adjusts its strategy based on emerging evidence, ensuring that the policies remain optimal under given constraints. By constructing an implicit model of the environment's dynamics, IB-MDP eliminates the need for a parameterized transition probabilities. Instead, the transition dynamics are inferred from historical data using a variance-normalized similarity metric. This method significantly reduces computational complexity while retaining the flexibility to refine decisions as new data become available.

A key feature of IB-MDP is its use of Monte Carlo Tree Search with Double Progressive Widening (MCTS-DPW), which enables efficient navigation through large state spaces without exhaustive data collection. This approach is particularly suited for experimental planning, where the goal is to balance exploration and exploitation in a computationally efficient manner. Furthermore, the framework incorporates a Bayesian sampling method, continuously refining the policy to incorporate new information, thus ensuring that the decision-making process adapts to changes in state uncertainty and target feature values over time.

To further enhance robustness and accuracy, IB-MDP integrates an ensemble method. By aggregating multiple policies generated from independent runs, the ensemble method mitigates inference bias and reduces variance, ensuring that the decision-making process is both reliable and adaptive. This combination of implicit modeling, dynamic policy adjustment, and ensemble learning offers a powerful tool for optimizing resource usage in complex experimental designs.

## 4.2 IB-MDP FORMULATION

The IB-MDP (Implicit Bayesian Markov Decision Process) framework can be defined as a tuple $\langle \mathcal{S}, \mathcal{A}, \mathcal{T}, \mathcal{R}, \gamma \rangle$, where:

- **States** ($\mathcal{S}$): The state space represents the knowledge about the drug candidate or system at each decision point.

- **Actions** ($\mathcal{A}$): A set of actions, where each action corresponds to selecting assays or experiments to perform.

- **Transition Function** ($\mathcal{T}$): Transition probabilities between states, implicitly derived from historical data $\mathcal{D}$ using a similarity-based metric.

- **Reward Function** ($\mathcal{R}$): The reward function that penalizes resource costs and rewards uncertainty reduction and goal achievement.

- **Discount Factor** ($\gamma$): A scalar discount factor that determines the present value of future rewards.

### 4.2.1 SIMILARITY WEIGHT FUNCTION

The transition function relies on a similarity weight $w_i(s)$ for each historical data point $D_{s_i}$. This is computed based on the distance between the current state $s$ and the historical data point $D_{s_i}$ using a variance-normalized distance metric: $w_i(s) = \exp\left(-\lambda_w \cdot d(s, D_{s_i})\right),$

where $\lambda_w$ is a scaling factor, and the distance metric $d(s, D_{s_i})$ is:

$$d(s, D_{s_i}) = \sum_{k=1}^{n} \lambda_k \cdot \frac{(s_k - (D_{s_i})_k)^2}{\sigma_k^2},$$

with $\lambda_k$ representing feature-specific scaling factors, and $\sigma_k^2$ being the variance of the $k$-th feature in $\mathcal{D}$.

### 4.2.2 IMPLICIT TRANSITION MODELING VIA SAMPLING

In the IB-MDP framework, the transition function $\mathcal{T}(s, a, s')$ is not explicitly defined through a known analytical function. Instead, it is implicitly modeled using a weighted sampling process that leverages historical data $\mathcal{D}$ and the similarity weights vector $W$. This process allows the transition

to a new state $s'$ to be based on past data points most similar to the current state $s$, according to a variance-normalized distance metric.

The transition probability from state $s$ to state $s'$ given action $a$ is defined as:

$$P(s'|s,a) = \sum_{i=1}^{N} \frac{w_i(s)}{\sum_{j=1}^{N} w_j(s)} \cdot \mathbb{I}[s' = s \oplus \Delta s(a, D_{s_i})],$$

where:

- $w_i(s)$ is the similarity weight between the current state $s$ and the historical data point $D_{s_i}$.
- $\Delta s(a, D_{s_i})$ is the change in state resulting from action $a$ applied to the historical data point $D_{s_i}$.
- $\oplus$ denotes the state update operation, combining the current state $s$ with the effect of action $a$ based on the sampled historical data.
- $\mathbb{I}[\cdot]$ is an indicator function that ensures the state update conforms to the sampled transition.
- $N$ denotes the total number of historical data points.

This transition is realized through a weighted sampling process. Specifically, a historical state $D_{s_{\text{sampled}}}$ is selected with probability proportional to its similarity weight $w_i(s)$. The sampling function $\delta(W, s)$ selects a data point $D_{s_{\text{sampled}}}$ from the historical dataset $\mathcal{D}$: $D_{s_{\text{sampled}}} = \delta(W, s) \cdot \mathcal{D}$, where $\delta(W, s)$ uses the similarity weights $W$ to sample a historical state $D_{s_i}$ from $\mathcal{D}$. Once the historical state is sampled, the new state $s'$ is updated as: $s' = \beta(s, \mathcal{D}, a) = s \oplus \Delta s(a, D_{s_{\text{sampled}}})$, where $\Delta s(a, D_{s_{\text{sampled}}})$ represents the change in state resulting from action $a$ applied to the sampled data point $D_{s_{\text{sampled}}}$. The $\beta$ function is considered as the implicit Bayesian update for the state via sampling. This update mechanism allows the system to dynamically evolve by incorporating the effects of historical actions, without needing an explicit transition function.

### 4.2.3 BAYESIAN UPDATE MECHANISM

After the transition to a new state $s'$, the similarity weights $W$ are updated based on the new state and historical data. The updated weights $W'$ are computed as: $W' = \text{update}(W, s', \mathcal{D})$ where the update mechanism reflects how the similarity weights are adjusted based on the new state $s'$, the action $a$, and the historical data $\mathcal{D}$.

This process dynamically adjusts the state transition based on the updated belief about the system, providing a probabilistic framework for modeling uncertainties.

### 4.2.4 REWARD FUNCTION

The reward function is defined to balance cost minimization and uncertainty reduction:

$$R(s,a) = \begin{cases} -\mathbf{c}(s,a) \cdot \boldsymbol{\lambda}, & \text{if } a \neq \text{eox}, \\ -M, & \text{if } a = \text{eox}, \end{cases}$$

where $\mathbf{c}(s,a)$ represents the cost vector for action $a$, $\boldsymbol{\lambda}$ is a vector of trade-off parameters, and $M$ is a large penalty for premature termination.

### 4.2.5 STATE UNCERTAINTY AND LIKELIHOOD

Uncertainty in state $s$, denoted as $\mathcal{H}(s)$, is computed as: $\mathcal{H}(s) = \frac{\sum_{i=1}^{N} w_i(s)(k_i - \bar{k}_w(s))^2}{\sum_{i=1}^{N} w_i(s)}$ where $k_i$ is the value of the target feature in $D_{s_i}$ and $\bar{k}_w(s)$ is the weighted mean of the feature: $\bar{k}_w(s) = \frac{\sum_{i=1}^{N} w_i(s) \cdot k_i}{\sum_{i=1}^{N} w_i(s)}$ The likelihood of achieving the desired outcome is computed as: $\mathcal{L}(s) = \frac{\sum_{i=1}^{N} w_i(s) \cdot \mathbb{I}[k_i \in [k_{\min}, k_{\max}]]}{\sum_{i=1}^{N} w_i(s)}$ where $k_i$ is the value of the target feature $k$ in the $i$-th historical data point, $w_i(s)$ is the similarity weight for $D_{s_i}$, and $\mathbb{I}$ is the indicator function. $k_{\min}$ and $k_{\max}$ are the user-defined lower and upper bound values of the target features.

### 4.2.6 TERMINAL CONDITION

The state $s$ is considered terminal when: $\mathcal{H}(s) \leq \epsilon$ and $\mathcal{L}(s) \geq \tau$ where $\epsilon$ is the state uncertainty threshold, and $\tau$ is the likelihood threshold.

## 4.3 THE IB-MDP ALGORITHM

The IB-MDP framework models sequential decision-making within resource-constrained environments, such as drug discovery, where an optimal sequence of experiments needs to be chosen under uncertainty. The action set at each state, represented as the power set of available assays $\mathcal{P}(A)$, may be constrained by the maximum number of assays $m$ (i.e., $\mathcal{A}_m$). Transitions between states are modeled using historical data and similarity weights, as outlined in the formulation section.

### 4.3.1 SOLVING IB-MDP WITH MCTS-DPW

To solve the IB-MDP problem, we employ Monte Carlo Tree Search (MCTS) with Double Progressive Widening (DPW). MCTS is a powerful search algorithm used to explore large state-action spaces by building a search tree through iterative simulations Browne et al. (2012). DPW is utilized to handle large and continuous action spaces by initially restricting the number of explored actions at each state and progressively widening the action set as more iterations are performed Couëtoux et al. (2011).

MCTS operates in four main steps: Selection, Expansion, Simulation, and Backpropagation. The Upper Confidence Bound (UCB) policy is used in the selection phase to balance exploration and exploitation: $a = \arg\max_{a' \in \mathcal{A}(s)} \left( Q(s, a') + c\sqrt{\frac{\ln N(s)}{N(s,a')}} \right)$ where $Q(s, a')$ is the estimated value of action $a'$ in state $s$, and $N(s)$ and $N(s, a')$ represent the number of visits to state $s$ and action $a'$ in state $s$, respectively.

The state transitions during the simulation phase are modeled implicitly via weighted sampling based on historical data, using the Bayesian update mechanism described earlier. This allows the IB-MDP to adapt dynamically to new information while maintaining computational efficiency.

### 4.3.2 PARETO FRONT GENERATION

After solving the IB-MDP problem and obtaining a set of optimal policies $\pi_j^*$ from the MCTS-DPW runs, the next step is to generate the Pareto front. This front helps to discern optimal trade-offs between competing objectives, such as minimizing cost and reducing state uncertainty.

The Pareto front consists of non-dominated points in the objective space, where each point represents a policy that is optimal under certain constraints. Mathematically, this can be formulated as: minimize $\{(\mathcal{C}(s), \mathcal{H}(s)) \mid s \in \mathcal{S}\}$ where $\mathcal{C}(s)$ is the cost associated with state $s$ and $\mathcal{H}(s)$ represents state uncertainty. A state $s'$ dominates state $s$ if: $\mathcal{H}(s') < \mathcal{H}(s)$ and $\mathcal{C}(s') < \mathcal{C}(s)$ indicating that $s'$ has lower uncertainty and lower cost. The Pareto front is the set of states that are not dominated by any other state, ensuring that each point on the front represents a trade-off between cost and uncertainty.

## 4.4 ENSEMBLE METHOD FOR IB-MDP

The ensemble method addresses the inherent variability in single runs of the IB-MDP algorithm, especially when using Monte Carlo Tree Search with Double Progressive Widening (MCTS-DPW). Stochasticity and sensitivity to initial conditions can lead to different Pareto fronts and optimal policies. By executing the IB-MDP algorithm $N$ times, each run generates an optimal policy $\pi_j^*$ and a corresponding Pareto front $\mathcal{P}_j$. Aggregating the results across multiple runs improves robustness, reduces bias, and enhances the reliability of decision-making.

**Advantages of Ensemble IB-MDP:** The ensemble IB-MDP methodology provides several advantages: **Improved Robustness:** By aggregating results from multiple simulations, the ensemble approach reduces the impact of variability and randomness in individual runs, enhancing the stability of decision outcomes. **Bias Reduction:** Exploring diverse decision trajectories across runs minimizes inference bias and yields more accurate estimates of optimal policies. **Predictive Power:** The

ensemble method helps identify patterns across multiple runs. The aggregation of these results informs the construction of a Maximum Likelihood Action Sets Path (MLASP), providing a candidate metric that can guide decision-making by assessing decisions under different likelihood thresholds $\tau$.

### 4.4.1 MAXIMUM LIKELIHOOD ACTION SETS PATH (MLASP)

The MLASP is a key outcome of the ensemble approach. It is constructed by identifying the most frequently occurring optimal action set at each uncertainty level $u$ across multiple runs. Specifically, for each uncertainty level, the action set $A_u^*$ that occurs most frequently is chosen as: $A_u^* = \arg\max_A \sum_{j=1}^N \mathbb{I}(A \in \mathcal{P}_j(u))$ where $\mathbb{I}$ is the indicator function that counts whether the action set $A$ is part of the Pareto front $\mathcal{P}_j(u)$ for the $j$-th run. By connecting all $A_u^*$ across varying uncertainty levels, we form the MLASP, ensuring robust decision-making across different scenarios.

Figure 2 illustrates the construction of the MLASP by showing a histogram of actions proposed across an ensemble of 50 independent runs of the IB-MDP algorithm. For a given state at the uncertainty threshold $\tau = 0.2$, the height of each bar represents the frequency with which a particular action was chosen by the ensemble. The action that appears most frequently across the runs for that threshold is considered the majority-voted action and is included in the MLASP.

The histogram provides a clear visual of how often each action was selected, and the bar with the highest frequency corresponds to the optimal action under the given uncertainty. The key insight from this figure is how the ensemble method ensures more robust and stable decision-making by leveraging majority voting across multiple runs. It demonstrates that, even with variability across different runs, the MLASP method converges on a reliable decision, enhancing both predictive power and robustness in the decision-making process.

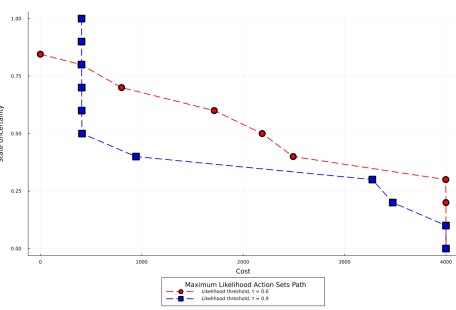

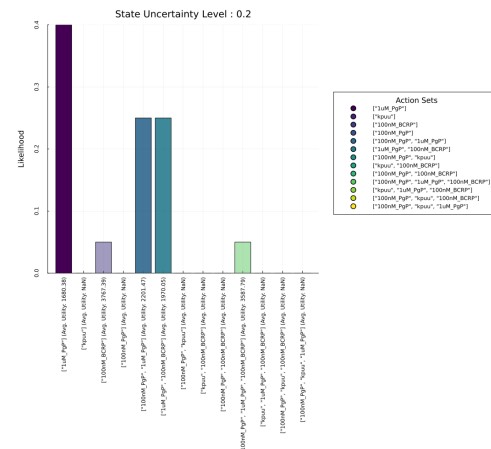

Figure 1: Exemplary Monetary-prioritized MLASP for two $\tau$ thresholds for the data point with 50 ensembles, $\text{QSAR}_{\text{mrt}}$ value of 1.56, $\text{QSAR}_{\text{100nM\_BCRP}}$ of 0.87, and $\text{QSAR}_{\text{1uM\_PgP}}$ of 0.513. Different $L$ likelihood result in different MLASP leading to distinct decision paths. A general trend is that the higher liklihood threshold $\tau$ value, the lower left MLASP will be.

Figure 2: Example of 50 ensemble IB-MDP proposed action in a histgram plot for the state uncertainty level = 0.2, and $\tau = 0.9$

### 4.5 ALGORITHM

For the detailed and complete description of the algorithm, see Algorithm 1.

## 5 EXPERIMENTS

### 5.1 EXPERIMENTAL SETUP

Our experimental setup utilizes a dataset of 220 compounds, each characterized by both *in silico* predictions and physical properties. The *in silico* features include Quantitative Structure-Activity

---

**Algorithm 1** Ensemble IB-MDP Algorithm

---

**Require:** Initial state $s_0$, historical data $\mathcal{D}$, similarity function $W$, Bayesian update function $\beta$, horizon $H$, number of iterations $n_{\text{itr}}$, number of ensemble runs $N$
**Ensure:** Pareto front of state uncertainty vs. expected utility costs
1: Initialize an array $\mathcal{P}$ to store Pareto fronts
2: **for** $j = 1$ to $N$ **do**
3:     Initialize MCTS-DPW tree with root node representing $s_0$
4:     **for** $i = 1$ to $n_{\text{itr}}$ **do**
5:         $s \leftarrow s_0$
6:         **while** not terminal and within horizon $H$ **do**
7:             Select action $a$ using UCB policy: $a = \arg\max_{a' \in \mathcal{A}(s)} Q(s, a') + c\sqrt{\frac{\ln N(s)}{N(s, a')}}$
8:             Simulate next state $s'$ using Bayesian update via sampling: $s' = \beta(s, \mathcal{D}, a)$
9:             Update similarity weights $W$ based on new state $s'$
10:           Update tree with $s'$ and reward $R(s, a)$
11:           $s \leftarrow s'$
12:         **end while**
13:         Backpropagate rewards and update $Q$ values along the path
14:     **end for**
15:     $\pi_j^* \leftarrow$ Extract optimal policy from tree
16:     $\mathcal{P}_j \leftarrow$ Compute Pareto front from $\pi_j^*$
17:     Append $\mathcal{P}_j$ to $\mathcal{P}$
18: **end for**
19: **for** each uncertainty level $u$ **do**
20:     $A_u^* = \arg\max_A \sum_{j=1}^N \mathbb{I}(A \in \mathcal{P}_j(u))$
21: **end for**
22: Construct Maximum Likelihood Action Sets Path (MLASP) from $A_u^*$
23: **return** MLASP

---

Relationship (QSAR) predictions, such as $\text{QSAR}_{\text{1uM\_PgP}}$, $\text{QSAR}_{\text{100nM\_BCRP}}$, and $\text{QSAR}_{\text{mrt}}$. In addition to these predictions, transporter activity data such as 100nM PgP, 1uM PgP, and 100nM BCRP are also considered. The financial and time costs associated with these transporter activities are estimated at \$400 per assay with a turnaround of 7 days, while $kpuu$ measurements incur a higher cost of \$4000 and take 21 days. These values highlight the substantial resource investment required for these tests.

To generate the Maximum Likelihood Action Sets Path (MLASP), we conduct up to three parallel assays, allowing simultaneous experimental operations. This setup helps reduce state uncertainty more efficiently while maximizing information gain, both of which are essential for effective decision-making. A computational threshold of 10 is applied to assess state uncertainty, ensuring that the algorithm captures meaningful differences in uncertainty levels.

We employ the IB-MDP algorithm, integrated with a Monte Carlo Tree Search (MCTS) solver using Double Progressive Widening (DPW). This solver runs for 20,000 iterations, with an exploration constant of 5.0, 50 ensembles, providing a balance between exploring new actions and exploiting known outcomes.

The primary goal is to identify the actions that achieve the greatest reduction in state uncertainty, comparable to performing the final target assay ($kpuu$), while minimizing both costs and resource use throughout the decision-making process.

**Experimental Computing Resources**:

We performed the IB-MDP simulations on an Apple M1 Pro chip with 16GB of memory. For each ensemble run, with 100 iterations of the IB-MDP per $\tau$ value, the estimated completion time was approximately 1 hour.

## 5.2 TRADITIONAL HEURISTIC DECISION RULES

The decision-making process for brain penetration assays typically relies on heuristic rules, primarily using QSAR (Quantitative Structure-Activity Relationship) predictions and the unbound brain-to-plasma partition coefficient (kpuu). These rules can be summarized as follows:

A compound is considered **promising** if: $QSAR_{1uM\_PgP} < 2$, $QSAR_{100nM\_BCRP} < 2$, and $0.5 \leq$ kpuu $\leq 1$. A compound is considered **non-promising** if either: $QSAR_{1uM\_PgP}$ or $QSAR_{100nM\_BCRP}$ exceeds 4, regardless of the kpuu value.

### 5.3 SELECTIVE CASE STUDY FOR COMPOUND SELECTION DECISION-MAKING

We tested the framework using three scenarios, designed to reflect different QSAR conditions. These case studies demonstrate the flexibility and robustness of our decision-making framework, showing its potential to improve the drug discovery process by identifying promising compounds that might be missed by traditional methods.:

**Baseline Confirmation**: This scenario tests compounds where both $QSAR_{1uM\_PgP}$ and $QSAR_{100nM\_BCRP}$ are below 2, and kpuu values fall within normal ranges. It serves to validate traditional decision-making processes.

**Heuristic Challenge**: In this scenario, compounds present borderline or conflicting QSAR data, with at least one QSAR value exceeding 4. This scenario tests the framework's ability to interpret complex signals and identify viable candidates.

**Opportunity Discovery**: This scenario evaluates compounds with high $QSAR_{1uM\_PgP}$ and $QSAR_{100nM\_BCRP}$ values, but acceptable kpuu. It aims to discover overlooked compounds that could be promising despite failing traditional heuristics.

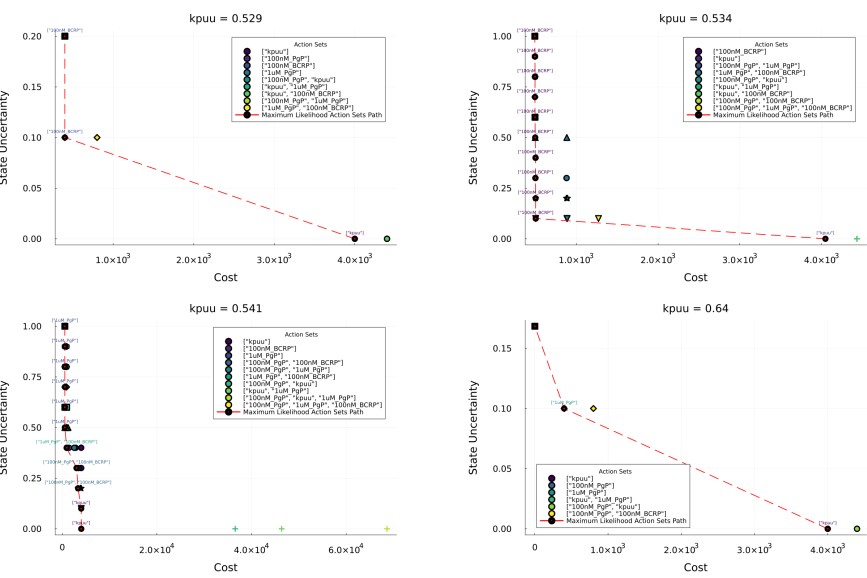

Figure 3: Monetary-prioritized IB-MDP results with MLASPs for four representative compounds, ordered by kpuu values to illustrate variations in QSAR metrics and corresponding recommended actions. For kpuu = 0.529, $QSAR_{1uM\_PgP} = 5.0$, $QSAR_{100nM\_BCRP} = 9.6$, and $QSAR_{mrt} = 0.99$. The IB-MDP recommends action is [100nM_BCRP]. For kpuu = 0.534, $QSAR_{1uM\_PgP} = 0.903$, $QSAR_{100nM\_BCRP} = 8.5$, and $QSAR_{mrt} = 2.64$. The recommended action is [100nM_BCRP]. For kpuu = 0.5407, $QSAR_{1uM\_PgP} = 1.68$, $QSAR_{100nM\_BCRP} = 1.3$, and $QSAR_{mrt} = 1.82$. The IB-MDP suggests actions are either [100nM_PgP, 100nM_BCRP] or [1uM_PgP, 100nM_BCRP]. For kpuu = 0.6400, $QSAR_{1uM\_PgP} = 21.4$, $QSAR_{100nM\_BCRP} = 0.73$, and $QSAR_{mrt} = 1.2$. Recommended actions include [1uM_PgP], indicating a high probability of effectiveness under the given experimental conditions.

### 5.4 EXPERIMENTAL RESULTS : COST COMPARISON BETWEEN CONVENTIONAL AND IB-MDP DECISIONS

The results of the IB-MDP exploration for the representative cases in Table 1 are shown in Figure 3. In the baseline scenario, the IB-MDP recommends actions involving [1uM_PgP, 100nM_BCRP],

[100nM_PgP, 100nM_BCRP], or 1uM_PgP, resulting in monetary costs ranging from \$400 to \$800, compared to the traditional cost of \$5200.

In the heuristic challenge scenario, the IB-MDP still proposes a single action along the MLASP with a \$400 cost, whereas traditional heuristic rules completely miss the opportunity to identify this promising compound. For the opportunity discovery scenario where all QSAR values are greater than 4, the IB-MDP successfully identifies a unique set of actions [100nM_PgP, 1uM_PgP] that significantly reduce state uncertainty. In contrast, the traditional rules fail to recognize this specific compound as a promising candidate.

Table 1: Comparison of Traditional Approach and IB-MDP Generated Costs for Selected Compounds

| $QSAR_{1uM\_PgP}$ | $QSAR_{100nM\_BCRP}$ | $QSAR_{mrt}$ | kpuu | 100nM_PgP | 1uM_PgP | 100nM_BCRP | Traditional Cost | IB-MDP Cost |
|---|---|---|---|---|---|---|---|---|
| 1.68 | 1.3 | 1.82 | 0.5407 | 1.06 | 0.79 | 1.32 | \$5200 | \$400 - \$800 |
| 0.903 | 8.5 | 2.64 | 0.5343 | 2.16 | 1.14 | 14.16 | \$5200 | \$400 |
| 21.4 | 0.73 | 1.2 | 0.6400 | 17.42 | 19.69 | 0.83 | \$5200 | \$400 - \$800 |
| 5.0 | 9.6 | 0.99 | 0.5289 | 15.92 | 12.86 | 8.23 | \$5200 | \$800 |

## 6 CONCLUSIONS

In this study, we present the Implicit Bayesian Markov Decision Process (IB-MDP), a framework designed to improve decision making under uncertainty in resource-constrained environments. By dynamically integrating historical data using a similarity-based metric, the IB-MDP refines beliefs about the current state in relation to target features. This refinement is achieved through implicit Bayesian updates with a sampling approach, ensuring the policy search maximizes information gain, minimizes costs, and achieves key experimental objectives within an optimal range.

A key advantage of the IB-MDP is its ability to significantly reduce state uncertainty without the need to perform the most expensive and resource-intensive assays, such as the target assay (kpuu). This not only enhances cost-efficiency but also accelerates decision-making by bypassing less critical, high-cost experiments.

Furthermore, IB-MDP benefits from an ensemble approach that aggregates policies across multiple runs, reducing variance, and improving robustness. By aligning maximum likelihood action sets with predefined probability bounds for target features, the methodology ensures consistent decision quality. Through its comprehensive and data-driven approach, the IB-MDP outperforms traditional heuristic methods, offering enhanced adaptability, precision, and resource optimization in experimental planning. It demonstrates significant potential in streamlining decision-making tasks in drug discovery and other fields requiring strategic resource management under uncertainty.

## 7 BROADER IMPACTS

The IB-MDP framework provides a versatile approach to adaptive decision making, offering benefits beyond preclinical assay scheduling. By integrating historical data, dynamic Bayesian updates, and ensemble methods, the framework enhances decision-making efficiency in various fields such as healthcare, logistics, and financial risk management. Its ability to handle uncertainty and resource constraints ensures robust, data-driven decisions, making it a valuable tool to improve outcomes in diverse industries.

## 8 LIMITATIONS

Increasing the number of runs $N$ enhances the accuracy and robustness of the optimal action set estimation but also increases computational costs. While the ensemble method helps reduce variance and bias, its efficiency decreases as $N$ grows, with diminishing returns typical in ensemble-based decision-making frameworks. The optimal value of $N$ depends on the problem's complexity and the available computational resources, as larger ensembles may be necessary to fully explore complex state-action spaces. In particularly intricate environments, more iterations may be required to ensure

reliable convergence, although the ensemble approach generally stabilizes after a sufficient number of runs.

Another important limitation lies in the framework's reliance on historical data. Although leveraging historical data helps to integrate similarity-based metrics for decision-making, it may not sufficiently account for novel scenarios in real-world experiments. Future extensions of the IB-MDP framework could incorporate more flexible strategies, such as adaptive kernel-based methods or deep learning approaches, to extrapolate to states not represented in the existing dataset.

Moreover, while thresholds for decision-making tend to converge toward a maximum likelihood action path, further exploration of how state uncertainty reduction influences the likelihood of achieving desired outcomes could offer additional insights. This might enable more effective ensemble adjustments, ultimately improving policy reliability and performance in dynamic environments.

## ACKNOWLEDGMENTS

We thank Karsten Menzel, Karin Otte, Kevin Bateman and Antong Chen for their helpful feedback and suggestions.

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
