# OpenReview forum: "Implicit Bayesian Markov Decision Process for Resource Efficient Decisions in Drug Discovery"
_ICLR.cc/2025/Conference — Submitted to ICLR 2025_

### Official Review · Reviewer_airy · 2024-11-01

**Soundness:** 1
**Presentation:** 2
**Contribution:** 2
**Rating:** 3
**Confidence:** 3

**Summary:**

The authors tackle the drug discovery problem, where researchers need to make sequential decisions to maximize the probability of success for drug candidates while minimizing expected costs. They developed an Implicit Bayesian Markov Decision Process (IB-MDP) algorithm, which constructs a model of the environment's dynamics using historical data. This algorithm also employs an ensemble approach to recommend actions that balance reducing uncertainty and optimizing costs. The effectiveness of IB-MDP is assessed through a drug discovery task.

**Strengths:**

The authors have applied the proposed method to a real drug discovery problem.

**Weaknesses:**

**Weakness 1: Related studies are not comprehensively investigated.**

Although not explicitly stated in the Abstract and Introduction, the author's problem seems to be formulated as a multi-objective reinforcement learning (MORL) problem from the description in Sec. 4.3.2.
Since there are many existing studies on MORL such as [Barrett 2008][Hayes 2022][Lu 2023], it is necessary to discuss the authors' formulation and method by citing related studies.

[Barrett 2008] Barrett, Leon, and Srini Narayanan. "Learning all optimal policies with multiple criteria." Proceedings of the 25th international conference on Machine learning. 2008

[Hayes 2022] Hayes, Conor F., et al. "A practical guide to multi-objective reinforcement learning and planning." Autonomous Agents and Multi-Agent Systems 36.1 (2022): 26.

[Lu 2023] Lu, Haoye, Daniel Herman, and Yaoliang Yu. "Multi-objective reinforcement learning: Convexity, stationarity and pareto optimality." The Eleventh International Conference on Learning Representations. 2023.

**Weakness 2: The design of the proposed method is unconvincing.**

The authors do not use a parametric model to estimate the transition function, but use the formula of Sec. 4.2.2, which is computed by sampling from historical data using the similarity weight function in Sec. 4.2.1. This approach does not appear to be theoretically justified. For example, if a sufficiently large amount of historical data exist, does this estimated transition function coincide with the true transition function?


**Weakness 3: Insufficient evaluation of the proposed method.**

The experiments seem to be limited to comparing the author's method with traditional heuristic decision rules in the drag discovery problem. I think it is necessary to show the method's superiority compared to existing RL methods, such as those cited in section 2 on some RL tasks.

**Questions:**

It would be helpful if the authors could provide a theoretical basis for the proposed method of estimating transition functions using similarity weights.

---

### Official Review · Reviewer_LLnj · 2024-11-02

**Soundness:** 3
**Presentation:** 1
**Contribution:** 2
**Rating:** 3
**Confidence:** 3

**Summary:**

The paper aims to formalize and tackle as a sequential decision-making problem the challenge of creating Research Operation Plans (ROPs) in the context of drug discovery, in particular for ADME studies. This problem, often tackled via rule-based heuristics or domain knowledge, is particularly challenging due to the complex nature of the objective function, lack of knowledge of the dynamics, and high-dimensionality of the state-action space. The authors formalize this problem as an MDP with unknown dynamics, propose an algorithm to tackle it, and perform an experimental evaluation against classic heuristics.

**Strengths:**

ORIGINALITY:
- the paper presents a new formalization of a pre-clinical experimental scheduling problem via a MDP formulation.
- the authors propose a combination of existing algorithmic ideas that seems to improve over classic techniques used.
- in Sec. 4.2 the authors present a way to sample transitions by leveraging offline historical data and a novel notion of distance

SIGNIFICANCE:
- the tackled problem is important and therefore any improvement over classic techniques used in the field could be particularly relevant. Nonetheless, I am not an expert in the specific applied area tackled within this paper and therefore cannot quantify the potential impact.

**Weaknesses:**

The paper contains explicit Acknowledgements, which given my understanding goes against ICLR policies as it can break double-blind reviewing. Nonetheless, I might be wrong about this.

ORIGINALITY and SIGNIFICANCE:
- the paper does not seem to bring any fundamental new idea from a RL/MDP viewpoint. As mentioned in the next paragraph, I believe that the paper should have been significantly more formal, clear and specific regarding the specific problem treated to highlight problem-specific contributions that deviate from existing RL methods.
- similarly, the experiments should show comparison with a naive RL baseline, as it is not fully clear the need/gain of a fairly complex algorithm.
- for ML/RL community it might be trivial that this problem can be casted as an MDP and likely be solved with existing methods. Therefore I believe there might be more applied venues (e.g., related with drug discovery) where the work's significance could be higher especially considering the first point in this list.

While the significance with respect to the specific application area might be good (as mentioned earlier I cannot evaluate it), the fundamental problem I find in this paper is listed in the following.

CLARITY and QUALITY:
I strongly believe the paper does not meet the quality and clarity of exposition expected for this conference. In particular, I note the following serious issues:
- this is an applied, problem-specific, paper. Nonetheless, the specific problem treated is mentioned multiple times in very broad and unclear terms, and is actually first introduced only within the Related Work section. It should instead be presented clearly from the abstract or introduction. In particular, it seems that in abstract and introduction the paper claims to contribute to a much broader area, while this becomes very specific later on.
- Sec. 3 formalizes the objective function of the sequential decision-making problem before even defining the mathematical spaces of the objects used (e.g., the state space), which is done only within the Algorithm section (Sec. 4.2). The structure of the problem should be defined within the problem setting section (i.e., Sec. 3) not later on within a method section. Moreover, the objective in Sec. 3 shows a reward function $R$, which seems to be not defined. Similarly, it states objects such as 'state uncertainty' $H(s)$, which, as well, is not defined.
- RL is a huge field and a vast array of methods has been already developed. When one wants to apply RL to a new field, they should (1) formalize the problem, (2) understand its challenges, (3) leverage existing methods if possible, and (4) if existing methods are not enough then one must develop a new method that tackles new (clear) challenges. Unfortunately, this paper fails in separating what has been already developed in RL and what methodological aspects are novel and relevant for the broader RL community.

**Questions:**

- Do the authors believe the work brings foundational RL methodology beyond the specific application?

As it seems that the main value of the paper lies in the practical impact for a specific problem, I suggest to center significantly more the paper on the explanation of the problem in details and restructure the presentation as mentioned above.

**Details Of Ethics Concerns:**

The paper contains explicit Acknowledgements, which given my understanding goes against ICLR policies as it can break double-blind reviewing. I might be wrong about this.

---

### Official Review · Reviewer_8198 · 2024-11-02

**Soundness:** 1
**Presentation:** 1
**Contribution:** 2
**Rating:** 3
**Confidence:** 3

**Summary:**

The authors propose an architecture for optimally defining a sequence of experiments to understand the likely effectiveness of a drug, where optimality is defined as a pareto frontier of minimal cost and high likelihood of success (of the drug). The architecture is a model-based RL approach. The authors conclude that the proposed architecture can define testing strategies that are cheaper and strategies that are not as aggressively exclusive of promising drug candidates.

**Strengths:**

Significance: The problem of optimally defining experimental procedures is highly important and thus, if the IB-MDP architecture proved affective, would be a significant contribution to the healthcare domain.
Quality: The MDP definition is reasonable and (to the best of my knowledge) reasonably aligns to the aims of recommending likely drug candidates at a minimal cost, as described in the paper (however, I have no experience in drug discovery). Additionally, the methodology of deriving a pereto frontier would be palatable to practioners wanting to use the model for deriving an optimal experimental protocol and thus the domain context has been well considered.

**Weaknesses:**

Notation, algorithm description and ease of reading
The authors leave multiple functions undefined including $\beta(s,\mathcal{D},a)$ and the update function in section 4.2.3, as such, it was impossible for me to confidently comment on the approach. Similarly, in section 5.1, the authors state a computational threshold of 10 was used however, I was unclear what this referred to.
Without clarifying the methodology in sections 4.2.2 and 4.2.3, it would be impossible for me to recommend this for publication (I have raised this in the questions below). That being said, I would encourage the authors more generally, to ensure that everything is explicitly defined in the paper.
For the proceeding, I interpreted the approach to be a non-parametric approach to model-based RL with search.

Experiment design and conclusions
Broadly speaking, the papers experimental protocol was quite vague with respect to what it was trying to show, I was unclear as to whether the authors intended to demonstrate that IB-MDP was the superior architecture for experiment design, or to demonstrate that experimental design was successfully solved (to the point of being used in the real world), by IB-MDP. In either case, the experimental design was lacking for the following reasons:
-	Assuming the aim was to demonstrate the superiority of IB-MDP: Whilst the authors mentioned that experimental design was a relatively underexplored area, in section 2 a number of architectures used in adjacent fields were mentioned. In order to demonstrate the superiority of IB-MDP, I would have expected a greater number of baselines to be used.
-	Assuming the aim was to demonstrate the superiority of IB-MDP OR to demonstrate that experimental design had been solved by IB-MDP: The results of the experiments were unclear and insufficient in the following ways:
o	The authors claim that the IB-MDP policies are no more than $800, despite all policies in Figure 3 recommending policies culminating in at least $4000. In addition, it is unclear how the figure of $5200 has been arrived at for the analysis;
o	The authors concluded that in contrast to IB-MDP “the traditional rules fail to recognize this specific compound as promising”. However, this conclusion is only reasonable if the compound did in fact turn out to be useful. More generally, I was assuming the analysis had been performed off-policy, in the sense that the model was not deployed in the real-world and thus was likely highly susceptible to erroneous generalisation errors i.e., assuming because the compound looked promising according to the model, doesn’t mean it was.

Literature understanding
Within sections 1 and 2, the authors broadly claim superiority of IB-MDP due to its non-parametric nature (i.e., “without the need for precise parameterization”). However, there exists an entire literature on model free RL methods which entirely side-step the issue of modelling the transition function, parametrically or non-parametrically.

**Questions:**

-	Please provide more explicit details on the mechanics defined in sections 4.2.2 and 4.2.3, in particular, defining the functions $\beta$ and how W is updated
-	Please confirm your intentions with the paper – do you intent to demonstrate the superiority of IB-MDP for solving experiment design OR the readiness of your proposed model for a real world application?
-	Please provide more details regarding how the monetary figures of the “traditional” policy and ID-MDP policy and please confirm why the ID-MDP figures reported in table 1 do not correspond with the figures in Figure 3.
-	Please confirm how the compounds described in sections 5.3 were selected – did all of these compounds turn out to be valid?

---

### Official Review · Reviewer_VShZ · 2024-11-03

**Soundness:** 2
**Presentation:** 1
**Contribution:** 2
**Rating:** 3
**Confidence:** 3

**Summary:**

The paper offers a formulation for decision-making-under-uncertainty problems common in drug discovery applications.
The authors then propose an algorithm based on Markov Decision Processes that balances between gaining information about the environment and minimizing cost during decision making.
Experiments are conducted to demonstrate the improvement over traditional, rule-based methods offered by the proposed solution.

**Strengths:**

The paper tackles an important problem of drug discovery.
The experiment section presents promising results showing the superior performance of the proposed method compared to baselines.

**Weaknesses:**

The paper's writing and organization could be improved.
Due to the page limit, many important components of the algorithm (the model, the Bayesian update rule, Monte Carlo tree search, the Pareto frontier) are discussed only in a cursory manner – I suggest expanding the sections corresponding to the most fundamental contributions and moving the rest to the appendix.

From my perspective, the biggest weakness of the paper is the lack of competitors in the experiment section.
On lines 95–97, the authors mention that Bayesian optimization (BayesOpt) methods are undesirable in their setting, which I don't think the authors have fully justified.
BayesOpt has found a lot of success in these sequential experimentation scenarios [1], and it's not quite clear what the authors mean by "such methods are often less effective" (than what?).
In fact, the proposed method shares many common ideas with BayesOpt (e.g., a similarity-based predictive model).

There are a number of hyperparameters that the authors could consider conducting ablation studies for.

The acknowledgement section should be removed.

[1] Garnett, Bayesian Optimization, Cambridge University Press 2023.

**Questions:**

How do the authors contrast their method with Bayesian optimization approaches?

---

### Meta-Review · Area_Chair_cgnF · 2024-12-20

**Metareview:**

This paper considers the problem of sequential decision-making under uncertainty for drug discovery by minimizing expected costs. The proposed approach is based on modeling the problem as a Markov Decision Process; learn the dynamics using historical data to select actions by balancing uncertainty and cost.

The reviewers' were in consensus about several critical weaknesss of this paper (summarized below). Authors' did not submit a rebuttal.
- The paper did not contextualize the problem and approach in the context of prior work.
- The proposed method is not well-motivated and described clearly.
- Experimental evaluation is not convincing and does not include several baselines.
- The writing of the paper needs significant improvement for both clarity and exposition.

For all the above reasons, I'm recommending to reject this paper and encourage the authors' to improve the paper based on the feedback from reviewers'.

**Additional Comments On Reviewer Discussion:**

There was no rebuttal from authors' and all reviewers' were in consensus.

---

### Decision · Program_Chairs · 2025-01-22

Reject